# The open-ocean missing backscattering is in the structural complexity of particles

Emanuele Organelli [1,2], Giorgio Dall'Olmo [1,3], Robert J.W. Brewin[1,3], Glen A. Tarran[1], Emmanuel Boss[4] & Annick Bricaud[2]

Marine microscopic particles profoundly impact global biogeochemical cycles, but our understanding of their dynamics is hindered by lack of observations. To fill this gap, optical backscattering measured by satellite sensors and in-situ autonomous platforms can be exploited. Unfortunately, these observations remain critically limited by an incomplete mechanistic understanding of what particles generate the backscattering signal. To achieve this understanding, optical models are employed. The simplest of these models—the homogeneous sphere—severely underestimates the measured backscattering and the missing signal has been attributed to submicron particles. This issue is known as the missing backscattering enigma. Here we show that a slightly more complex optical model—the coated sphere—can predict the measured backscattering and suggests that most of the signal comes from particles >1 μm. These findings were confirmed by independent size-fractionation experiments. Our results demonstrate that the structural complexity of particles is critical to understand open-ocean backscattering and contribute to solving the enigma.

[1] Plymouth Marine Laboratory, Prospect Place, The Hoe, PL1 3DH Plymouth, UK. [2] Sorbonne Université, CNRS, Laboratoire d'Océanographie de Villefranche, LOV, F-06230 Villefranche-sur-Mer, France. [3] National Centre for Earth Observation, Plymouth Marine Laboratory, Prospect Place, The Hoe, PL1 3DH Plymouth, UK. [4] School of Marine Sciences, University of Maine, Orono 04469 ME, USA. These authors contributed equally: Emanuele Organelli, Giorgio Dall'Olmo. Correspondence and requests for materials should be addressed to E.O. (email: emanuele.organelli@obs-vlfr.fr)

Microscopic organic particles in the open ocean support a suite of processes, known as the biological carbon pump, which is ultimately responsible for transferring atmospheric $CO_2$ from the sunlit surface to the abyss. The biological pump thus modulates atmospheric $CO_2$[1,2] and supports deep water ecosystems and the fisheries upon which humans depend. Understanding this pump is of the utmost importance.

To achieve this understanding, measurements of the light scattered by marine particles are crucial[3–9]. Optical scattering is quantified by the volume scattering function (VSF) that measures the fraction of incident light deflected by particles in a given direction per unit distance[10]. By integrating the VSF in the forward and backward directions, one obtains the forward- and backscattering coefficients that are the most commonly measured optical proxies for particle concentration. Although the particulate backscattering coefficient ($b_{bp}$) is only a small fraction (1–2%) of the total scattering (or of the particulate beam attenuation coefficient $c_p$[11]), it has the great advantage of being directly linked to space-based measurements[12]. In addition, $b_{bp}$ can be measured in situ by miniaturised instruments installed on autonomous robotic platforms that sample at high vertical and temporal resolution from the surface to the mesopelagic zone (200–1000 m)[13]. Thus, optical backscattering measurements can be exploited to better understand the biological carbon pump.

The particulate optical backscattering coefficient can be used to estimate, through empirical relationships, the concentration of particulate organic carbon (POC)[14–16] and phytoplankton carbon[17,18], from which algal physiological parameters can be derived[19]. The $b_{bp}$ spectrum provides a proxy of the particle size[20] and can be used to analyse the phytoplankton community structure and derive group-specific carbon biomass[21]. Yet, the biogeochemical quantities derived from these algorithms suffer from large uncertainties. For example, the performance of POC and phytoplankton carbon empirical models can vary widely[15–18].

Difficulties in interpreting $b_{bp}$ arise due to an incomplete mechanistic understanding of which particles are detected by $b_{bp}$. The $b_{bp}$ coefficient has been mainly modelled using Mie theory[22–25] assuming that marine particles can be represented as homogeneous spheres. Although this assumption is sufficient to model the forward scattering of particles[25–27], this approach severely underestimates the phytoplankton backscattering[28–30] and the particulate backscattering in the open ocean[31]. This mismatch between theory and observations is known as the missing backscattering enigma[32]. To explain this missing backscattering, submicron detrital particles have been suggested as the major source of oceanic particulate backscattering[26,27,33,34]. Hence, it is currently assumed by many that particulate forward- and backscattering coefficients detect different oceanic particles.

Alternative approaches have been proposed to model marine particles as heterogeneous coated spheres[35–42] or as non-spherical particles[42–46]. Heterogeneous coated spheres represent the simplest way to model the bulk structural complexity of marine particle populations. Coated spheres can reasonably represent the external and internal cellular structures (e.g., cell wall and chloroplasts) of marine spherical and non-spherical phytoplankton, and account for differences in their chemical nature[35–42]. Coated spheres can also be used to simulate aggregates of living and detrital particles[47,48] and algal colonies[49]. Coated-sphere models produce backscattering signals that can be an order of magnitude higher than those predicted by Mie theory for the same overall diameter and volume-averaged particle composition (i.e., bulk refractive index[38]). Coated models, however, require parameters that quantify the thickness and chemical composition of each part of the particle and the choice of these parameters is critical[32]. Non-homogeneous spherical models thus have not been applied to study oceanic backscattering due to the high variability in the size and nature of natural particles. Prudence has been suggested for these more complex approaches[32].

Here we show that the open-ocean missing backscattering can be found in the structural complexity of marine particles. We have applied a coated-sphere model to estimate the backscattering of particle populations across the Atlantic Ocean and through mid-ocean gyres (Supplementary Fig. 1). We based our analysis on coincident measurements of particle size distributions (PSDs; 0.59–60 μm) and optical measurements at the wavelength 532 nm (see Methods). We performed the analysis only at 532 nm because it allows us to minimise the second-order effect of particulate light absorption on scattering[25], as well as to directly compare our results with previous results based on the homogeneous-sphere assumption[27]. We then examined the contributions to modelled scattering of different size fractions and confirmed the findings by additional optical measurements collected through independent size-fractionation experiments. Our results show that particles larger than 1 μm contribute the majority of the $b_{bp}$ measured across the Atlantic Ocean and that backward and forward scattering therefore detect different characteristics of particles within similar size ranges. We anticipate our results will open the door to a new way of interpreting marine optical backscattering measurements.

## Results

**Homogeneous spherical particles cannot entirely explain $b_{bp}$.** The homogeneous-sphere model (Mie theory) could not reproduce simultaneously the measured particulate beam attenuation ($c_p$) and backscattering ($b_{bp}$) coefficients (Fig. 1). Modelled $c_p$ and $b_{bp}$ at the ocean surface and at the level of the deep chlorophyll maximum (DCM) were tightly correlated with measured values regardless of the refractive index ($n$) assumed (Fig. 1; Supplementary Table 1). However, biases were evident and varied with $n$. When $n$ was set equal to 1.06, the homogeneous-sphere model accurately reproduced $c_p$ (Fig. 1a) but severely underestimated $b_{bp}$ (Fig. 1b). Modelled $b_{bp}$ values could be forced to match observations by increasing $n$ to 1.11 (Fig. 1d), but this increase in $n$ enhanced the bias in $c_p$ predictions by about 10 times (Fig. 1c). These results were robust to variations in the refractive index within the PSD (Supplementary Fig. 2), as well as to the presence of absorbing particles (Supplementary Fig. 3). The accurate prediction of $c_p$ obtained for $n$ equal to 1.06 suggested that Atlantic open-ocean particle populations were composed by phytoplankton-like organic material[50].

**A coated-sphere model for open-ocean particles.** In contrast to the results obtained using Mie theory, both $c_p$ and $b_{bp}$ coefficients could be accurately and simultaneously reproduced by a coated-sphere model (Fig. 2). No missing backscattering was observed for various combinations of the thickness of the outer coat ($tk_2$) with assigned refractive index ($n_2$) (Fig. 2b). The ensemble of model parameterisations was selected among the $tk_2$–$n_2$ combinations that returned the lowest errors in $b_{bp}$ prediction (Fig. 3, Supplementary Fig. 4). This selection was achieved by excluding $tk_2$–$n_2$ combinations that yielded values of the refractive index of the core of the sphere ($n_1$) too low for marine particles (i.e., $n_1 < 1.02$)[26,27,51] or unrealistic ($n_1 < 1$). Combinations with $n_2$ equal to 1.22 were also disregarded because they were too high for algal cellular constituents[50]. The selected parameterisations thus included 12 combinations of $4\% \le tk_2 \le 15\%$ and $1.12 \le n_2 \le 1.20$, with $n_2$ decreasing as a function of $tk_2$ according to a power law (Fig. 3). Corresponding values of $n_1$ varied between 1.022 and 1.042 (Fig. 3). These 12 combinations had $n_2$ values systematically higher than for $n_1$, which implies that the outer coat of the spheres must always be composed of more refracting matter than the inner core. The shapes of the VSFs we obtained from these selected parameterisations were similar to existing measurements (Supplementary Fig. 5).

**The coated-sphere model is validated with independent data.** The selected parameterisations could also reproduce field $c_p$ and $b_{bp}$ coefficients for an independent dataset (Fig. 4). These surface data were also collected across the Atlantic Ocean (Supplementary

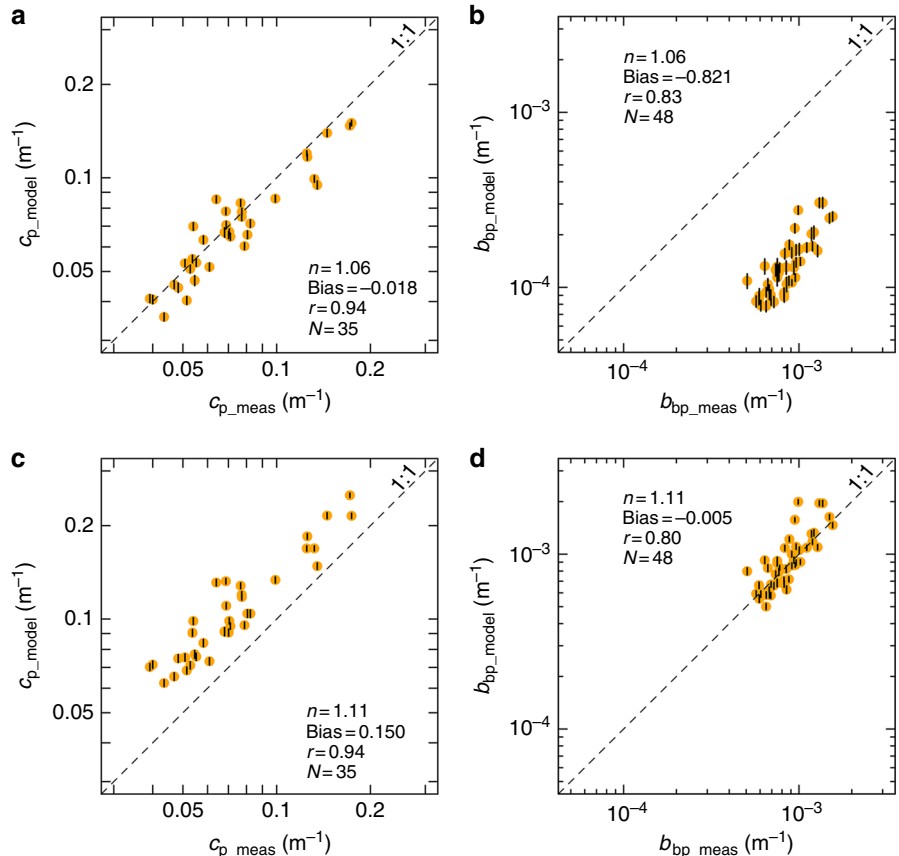

**Fig. 1** Mie theory cannot simultaneously reproduce beam attenuation and backscattering coefficients. Modelled vs. measured particulate beam attenuation (**a**, **c**) and optical backscattering (**b**, **d**) coefficients at 532 nm. Comparisons are based on data collected at the ocean surface (5 m) and at the DCM level during the AMT26 cruise. **a** and **b** indicate model results with refractive index $n$ equal to 1.06; **c** and **d** for $n$ equal to 1.11. Systematic error (Bias), Pearson's correlation coefficient ($r$) and number of observations ($N$) are shown. All $r$ coefficients are statistically significant ($p < 0.01$). Error bars (black vertical lines within each point) represent the combined uncertainty (95% confidence intervals) as propagated from particle size distribution measurements (see Methods)

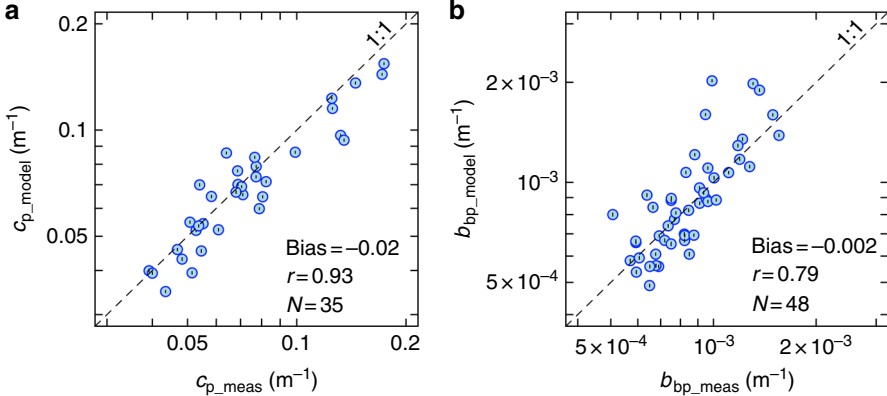

**Fig. 2** The coated-sphere model can simultaneously predict both beam attenuation and backscattering coefficients. Modelled vs. measured particulate beam attenuation (**a**) and optical backscattering (**b**) coefficients at 532 nm. Estimated values are average predictions derived from the selected ensemble of parameterisations. Error bars (black lines within circles) represent the combined uncertainty (95% confidence intervals) as propagated from particle size distribution measurements (see Methods). Systematic error (Bias), Pearson's correlation coefficient ($r$) and number of observations ($N$) are shown. All $r$ coefficients are statistically significant ($p < 0.01$). Data were collected at the ocean surface (5 m) and at the DCM level during the AMT26 cruise

Fig. 1) but covered a broader range of $c_p$ and $b_{bp}$ values than the one used for model training. Outside of the range of values used for the model parameterisation (i.e., $b_{bp}(532) > 1.56 \times 10^{-3}$ m$^{-1}$), $b_{bp}$ predictions were less accurate (Fig. 4b). This suggests that the particulate optical backscattering may be more sensitive than $c_p$ to changes in particle composition and structural characteristics.

**$c_p$ and $b_{bp}$ detect particles within similar size ranges.** The selected coated-sphere parameterisations predict that both the beam attenuation and backscattering coefficients measured across the Atlantic Ocean are generated by particles in approximately the same size range (Fig. 5; Supplementary Fig. 6). In agreement with previous studies[23,27], 90% of $c_p$ was generated from

particles with diameters between 0.59 and 7 μm, and <10% was produced by measured submicron particles. However, in stark contrast to previous studies based on the homogeneous-sphere model, when using the coated-sphere model we found that 90% of the backscattering signal came from particles between approximately 0.59 and 10 μm, whereas submicron particles between 0.59 and 1 μm contributed <20% of the signal (Fig. 5b).

**Independent confirmation of our findings**. The smaller-than-expected contribution of submicron particles to surface optical backscattering was independently confirmed by additional measurements of 1-μm filtered water samples, which also accounted for particles with diameters <0.59 μm (Fig. 6). Tests on the efficiency of these filtrations indicated that the filter retained the majority of particles for diameters closest to and above its nominal pore size limit (i.e., 1 μm), whereas progressively increasing amounts of particles with diameters smaller than 1 μm

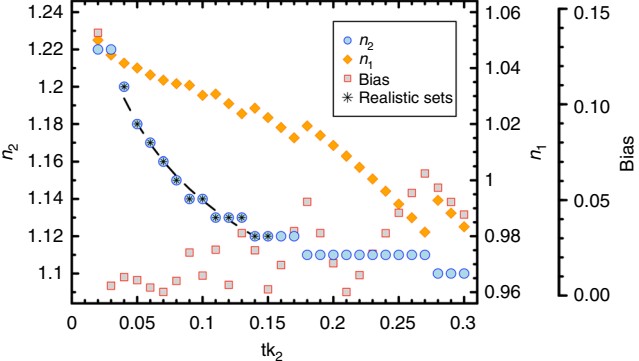

**Fig. 3** Realistic sets of coated-sphere model parameters. Combinations of coat thickness ($tk_2$) and refractive index ($n_2$) that minimise the systematic error (Bias) of predictions for $b_{bp}$ coefficients at 532 nm using a coated-sphere model with respect to in-situ optical measurements. $tk_2$ is expressed as a fraction of the radius of the modelled spherical particle. The volume-averaged refractive index $n$ is set equal to 1.06. Variations of the refractive index of the core ($n_1$) as a function of $tk_2$ and $n_2$ are also shown. Black stars indicate the 12 realistic combinations of coated-model parameters selected for particle populations at the ocean surface (5 m) and at the DCM level. The dashed line is the fit to the realistic sets of coated-sphere parameters and corresponds to $n_2 = 1.013\ (\pm 0.003)\ tk_2^{-0.051\ (\pm 0.002)}$ ($r = 0.99$, $p < 0.01$, $N = 12$)

passed through the filter (Fig. 6a). These measurements demonstrated that even when 90 ± 10% of particles with diameters larger than 1 μm were removed (Fig. 6a), on average only 40 ± 6% of the measured $b_{bp}$ signal was generated by the submicron particles (Fig. 6b). Our size-fractionation experiment further confirmed that particles < 1 μm contributed on average 20 ± 2% of the $c_p$ signal[23,27]. These size-fractionation results are also in agreement with previous studies that suggested that the $b_{bp}$ generated by particles with diameters smaller than 0.2 μm is negligible[31,52].

**The coated-sphere model reproduces in-situ $b_{bp}$-to-$c_p$ ratios**. The $b_{bp}$-to-$c_p$ ratio approximates, when the light absorption by particles is negligible (see Methods), to the ratio between $b_{bp}$ and total scattering (i.e., backscattering ratio), which measures the efficiency with which light is backscattered by particles[25]. The $b_{bp}$-to-$c_p$ ratios for a coated sphere can be 40 times higher than for homogeneous spheres for particles with diameters between 1 and 12 μm, and 8 times for submicron particles (Fig. 7a). Estimated ratios using the coated-sphere model reproduced the ratios measured in situ. For the samples used to parameterise and validate the coated-sphere model, the modelled median value was 1.27 ± 0.09% (N = 125). This modelled value was consistent with those derived from our (1.27 ± 0.30%, N = 125; Fig. 7b) and other field optical measurements[14,31,53], though with a standard deviation significantly smaller than observed in situ. This agreement between model results and experimental observations further highlights that, at the very least, a coated-sphere model is needed to reproduce oceanic $b_{bp}$.

**Discussion**
Experimental[28–30,42] and theoretical[43–45] studies have demonstrated that the optical backscattering of marine phytoplankton is severely underestimated when cells are modelled as homogeneous spheres. The homogeneous-sphere model also underestimates the optical backscattering generated by phytoplankton and other marine organic particles in the open ocean[31]. Here we showed that by modelling open-ocean particles as a population of coated spheres, we simultaneously predicted the measured beam attenuation and backscattering coefficients with considerably smaller biases than those obtained using the homogeneous-sphere model, over a wide range of trophic conditions across the sunlit Atlantic Ocean. We can thus provide an interpretation of in-situ oceanic particulate backscattering measurements that is not based on the homogeneous-sphere model. In contrast with previous interpretations based on the homogeneous-sphere model, our theoretical and experimental results suggest that the majority of

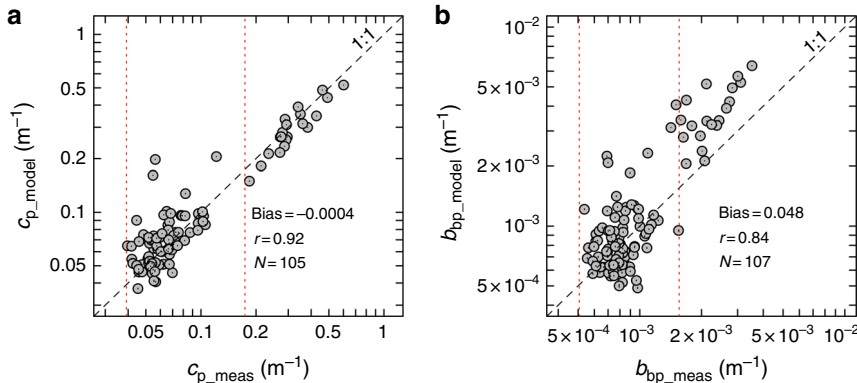

**Fig. 4** Independent validation of the coated-sphere model. Modelled vs. measured particulate beam attenuation (**a**) and optical backscattering (**b**) coefficients at 532 nm for an independent validation dataset. This dataset includes only samples collected at 5 m from the ship's underway seawater supply during the AMT22 cruise. Error bars (black lines within circles) represent the combined uncertainty (95% confidence intervals) as propagated from particle size distribution measurements (see Methods). Systematic error (Bias), Pearson's correlation coefficient ($r$) and number of observations ($N$) are shown. All $r$ coefficients are statistically significant ($p < 0.01$). Red dotted lines indicate the range of measured values of the particulate beam attenuation and optical backscattering coefficients used to parameterise the coated-sphere model

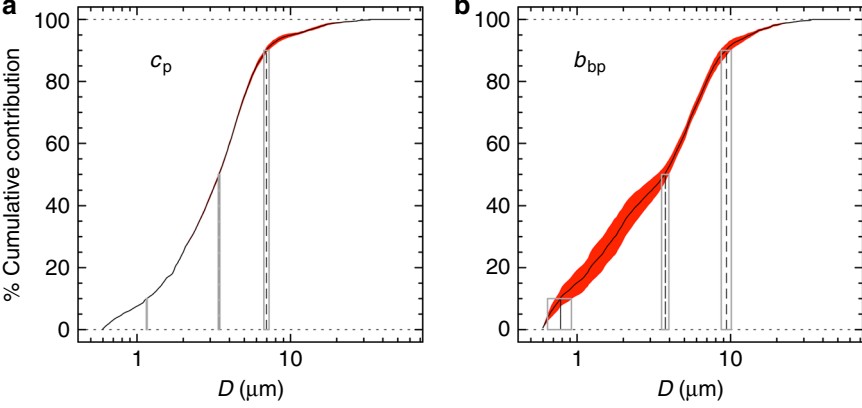

**Fig. 5** Beam attenuation and backscattering coefficients detect particles of similar sizes. Cumulative percent contributions (units of %) for modelled particulate beam attenuation (**a**) and optical backscattering (**b**) coefficients at 532 nm, as a function of the particle diameter ($D$, units of μm). Black solid line indicates the median value and red areas represent the standard deviation obtained using the selected coated-sphere parameterisations. Vertical rectangles correspond to the median ± 1 standard deviation of the size thresholds of particle diameters causing 10%, 50%, and 90% of the signal, from left to right of each plot, respectively. This analysis is based on data collected during the AMT26 cruise at the ocean surface (5 m) and at the DCM level

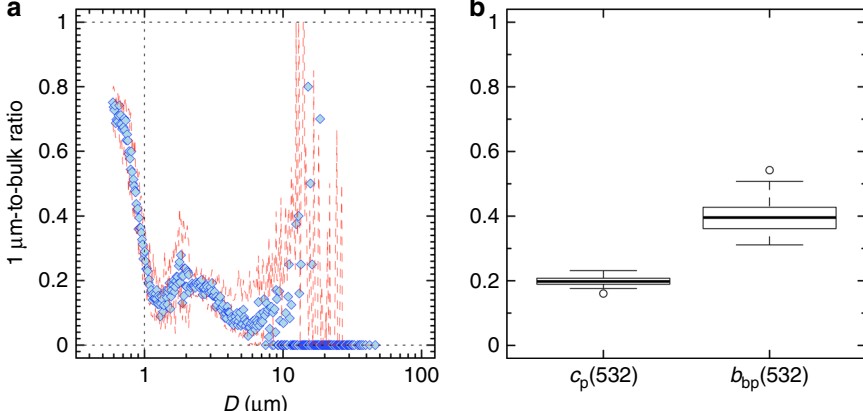

**Fig. 6** The optical contribution of submicron particles after 1 μm filtration. **a** Ratio of particle size distributions measured after and before filtration of 5-m water samples through a 1-μm filter as a function of the particle diameter ($D$, units of μm). Each point is the median of all ratios for samples collected during the AMT26 cruise ($N = 22$; Supplementary Fig. 1) for a given particle diameter. Red dashed lines represent the first and third quartiles. **b** Boxplot for the corresponding 1μm-to-bulk ratios of $c_p$ and $b_{bp}$ coefficients at 532 nm. The thick black line indicates the median value. Upper and lower side of each box are the third and first quartiles, respectively. Vertical bars represent minimum and maximum values. Open circles are outliers

the particulate backscattering coefficient in the studied area is due to particles with equivalent diameters between 1 and 10 μm, and that submicron particles generate <40% of the measured signal (Figs. 5b, 6b). This contribution by submicron particles is half that predicted by existing modelling studies for homogeneous spheres with theoretical power-law PSDs with an exponent of −4 and having a low refractive index (i.e., 1.05)[27]. Our modelling results may have underestimated the contribution of submicron particles because we used an integration limit of 0.59 μm for measured PSDs[23]. However, independent size-fractionation experiments confirmed the reduced optical contribution for all submicron particles (i.e., 0.001–1 μm; Fig. 6b) regardless of the lower size limit of our PSDs. According to our results, therefore, beam attenuation and backscattering coefficients most likely sense different characteristics of particles within similar size ranges (Fig. 5). Consistent with published results, the beam attenuation and thus forward scattering are sensitive to the average characteristics of particles[54]. Backscattering, on the other hand, reflects the characteristics of the outer shell of the coated sphere[35] or, in more general terms, of the particle structure. We consider the coated sphere as a simple but efficient way to model the structural complexity of marine particles[36,41]. A coated sphere may account for the effects on the optical backscattering of

external and internal phytoplankton cell structures such as the cell wall, cell membrane, chloroplasts, and other organelles[35,36,39–41]. The coated sphere may also reasonably reproduce the scattering of quasi-spherical organisms[36,42] and can be used to model the optical properties of amorphous agglomerations of many particles[47,48] or algal colonies[49]. Thus, the coated sphere can account for various levels of structural complexity that characterise particles in the marine environment. More in-depth studies are needed to identify exactly what particle characteristics are responsible for the backscattering effects reproduced by the coated sphere.

To understand what might be the consequences of using the homogeneous-sphere model to simulate the scattering properties of complex natural particles (e.g., phytoplankton), it is instructive to attempt to model the scattering of a complex particle by only using homogeneous spheres. Figure 8 demonstrates that, to reproduce the VSF of a coated sphere, multiple homogeneous particles of different sizes are needed. Specifically, as expected from previous studies[25], one homogeneous sphere of the same size and volume-averaged refractive index as the coated sphere can reproduce the forward scattering. However, to also reproduce the backscattering of the coated sphere, we must add the scattering intensities of multiple small spheres to that of the largest

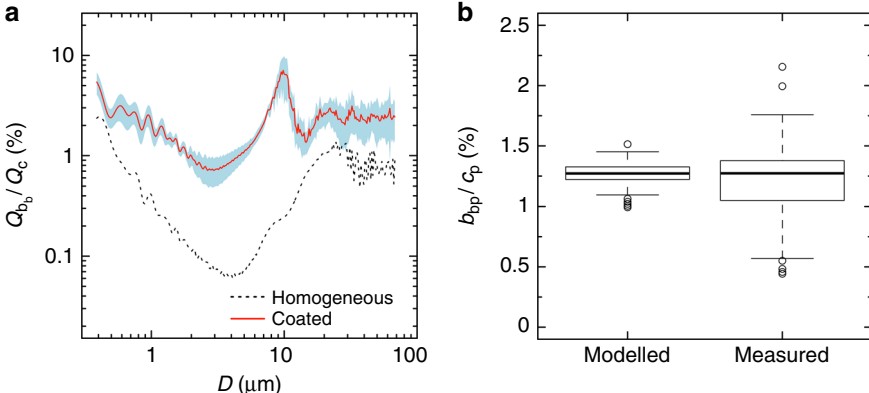

**Fig. 7** Modelled and measured $b_{bp}$-to-$c_p$ ratios. **a** Percent ratios of particulate backscattering ($Q_{b_b}$) to the beam attenuation ($Q_c$) efficiency factors at 532 nm, as a function of the particle diameter ($D$, units of µm). $Q_c$ is corrected for the acceptance angle effect (see Methods). The red solid line is the average ratio for the selected coated-sphere parameterisations and blue areas are the standard deviation. The black dashed line indicates the backscattering ratio for homogeneous spheres with equivalent volume-averaged refractive index. **b** Boxplot of percent ratios between particulate optical backscattering ($b_{bp}$) and beam attenuation ($c_p$) coefficients at 532 nm, as estimated using the coated-sphere model ensemble and measured in situ (AMT22 and AMT26 data). The thick black line indicates the median value. Upper and lower side of each box are the third and first quartiles, respectively. Vertical bars represent minimum and maximum values. Open circles are outliers

sphere, as they are the only homogeneous particles with a high backscattering efficiency[25]. We thus postulate that previous conclusions regarding the sources of oceanic backscattering based on the homogeneous-sphere model might have been distorted by the same limitations illustrated in the example of Fig. 8. In summary, an optical model that is too simplistic to represent the complexity of marine particles might have considerably altered our interpretation of the sources of oceanic backscattering.

Some marine particles might, however, be represented by homogeneous models[45,46] that could modify the contribution to the optical backscattering by the different size fractions. In our analysis, we assumed all marine particles to be coated spheres with similar characteristics because we did not have data to distinguish coated from homogeneous particles and to characterise their morphology. However, we performed a sensitivity analysis to understand how many homogeneous and coated spherical particles within the same population could still accurately reproduce the in-situ backscattering measurements. We found that modelled backscattering coefficients reproduced measured values until homogeneous particles represented up to 25% of the total number of particles measured (Supplementary Fig. 7). We therefore conclude that at least 75% of the marine particles sampled were too morphologically or structurally complex to be represented as homogeneous spheres. These complex spheres likely also represented agglomerations of both living and detrital material[47–49], in addition to freely dispersed individual particles[36,41,42].

The coated-sphere model that was parameterised using surface data underestimated the backscattering in the mesopelagic region of the Atlantic Ocean. Figure 9 shows that the selected parameterisations for the coated-sphere model predicted 61 ± 20% of the measured backscattering in the mesopelagic zone. Although in-situ $b_{bp}$ measurements in the clearer mesopelagic waters may suffer from higher uncertainties[55], the differences between modelled and measured backscattering coefficients could be due to particles with thicker and harder coats or to denser and more structurally complex aggregates. Moreover, the volume-averaged refractive index of mesopelagic particles may be different from the 1.06 value found for surface particles. For example, a few modelled beam attenuation coefficients at 532 nm for particles collected between 150 and 200 m ($N = 15$) were accurately predicted when the volume-averaged refractive index was set equal to 1.07 (Bias = −0.01).

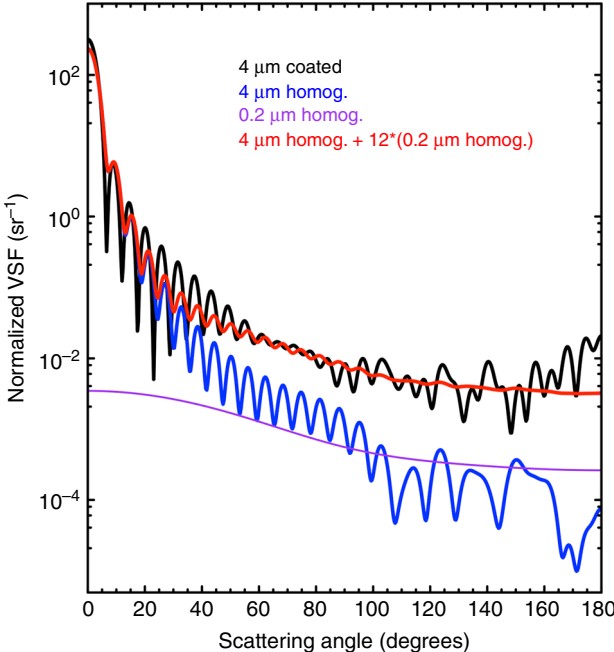

**Fig. 8** The homogeneous-sphere model can only reproduce the shape of the volume scattering function (VSF) of a complex particle by using multiple homogeneous spheres. Only the forward scattering (angles < 90°) of a coated sphere (black line) can be reproduced by a homogeneous sphere (blue line) with the same size and volume-averaged refractive index. To reproduce the backscattering (angles > 90°) of the coated sphere, twelve homogeneous spheres with diameter equal to 0.2 µm purple line and the same volume-averaged refractive index of the coated sphere must be added to the largest homogeneous sphere (red line). The refractive index of the coat of the coated sphere was set equal to 1.13, the coat thickness was 13% of the radius and the volume-averaged refractive index of the whole particle was set equal to 1.06. The backscattering ratios are equal to 0.88% and 0.85% for coated and combined homogeneous spheres, respectively. VSFs are normalised to the total scattering, and are shown for the wavelength 532 nm

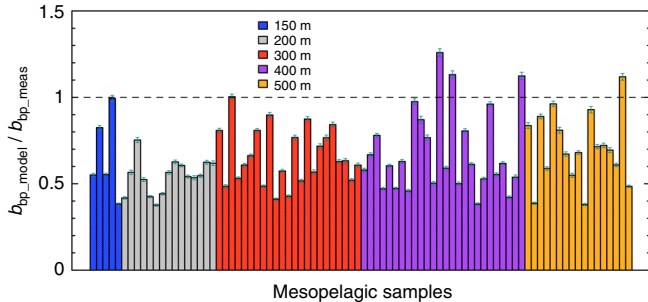

**Fig. 9** The backscattering coefficient is underestimated in the mesopelagic zone. Each bar indicates the average ratio between modelled optical backscattering coefficient at 532 nm derived from the selected coated-sphere parameterisations to that measured for a given sample in the mesopelagic zone, as collected along the AMT26 transect. Error bars represent the combined uncertainty (95% confidence intervals) as propagated from particle size distribution measurements (see Methods). Dashed line indicates the 1:1 ratio

A coated-sphere model might therefore disclose additional information on particle characteristics from optical data. This hypothesis is also supported by the higher uncertainties in the modelled backscattering coefficients that were found for the AMT22 samples when $b_{bp}$ was higher than $1.56 \times 10^{-3}\,\text{m}^{-1}$ (Fig. 4b). These AMT22 data were collected in surface waters that were not sampled in the AMT26 dataset used to parameterise the coated-sphere model. These waters were generally rich in large cells such as diatoms, as confirmed by the two-time higher relative pigment contribution than observed along the AMT26 transect (Supplementary Fig. 8). Algal populations encountered on AMT26 were mainly characterised by cyanobacteria and picoeukaryotes mixed with some coccolithophores, cryptophytes and other nanoeukaryotes (Supplementary Fig. 9). Thus, the observed vertical and regional differences predicted by the coated-sphere model suggest that oceanic $b_{bp}$ might be better estimated by parameterising the thickness and composition of the coat according to depth and trophic regime. In turn, we hypothesise that if this variability in the parameterisation could be inferred from combinations of optical measurements (e.g., $b_{bp}/c_p$), then new information of particle characteristics might be extracted from optical measurements. This new information could also help explain why predictions of POC are less noisy when using algorithms based on $c_p$ rather than on $b_{bp}$[14,16], and why performance of empirical algorithms for phytoplankton carbon varies widely[17,18].

Understanding which particle characteristics determine optical scattering coefficients in natural waters may also shed light on why community production is better predicted from $c_p$ than $b_{bp}$[56]. Particle and phytoplankton growth and loss rates can be derived from $c_p$ diel cycles[57], and the correlation between $c_p$ and $b_{bp}$ observed in oceanic waters[31,53,58] may suggest the same retrievals could also be possible by satellite observations of $b_{bp}$. However, $c_p$ and $b_{bp}$ diel cycles are out of phase[59] and diel phytoplankton carbon-specific backscattering and attenuation coefficients exhibit different behaviours[60]. Understanding the contribution of phytoplankton to in-situ measurements of $c_p$ and $b_{bp}$ might thus increase our capability to predict oceanic carbon biogeochemical rates. We therefore suggest that a potential new avenue of research could be to identify what changes in cell ultrastructures and composition govern diel variations of optical scattering properties.

The coated-sphere model predictions compared favourably to in-situ observations only when a specific set of parameters was selected. These selected parameters represent the known metabolite composition of oceanic particles[50]. The volume-averaged refractive index equal to 1.06 reveals that particle populations across the Atlantic Ocean are mainly of organic origin[50]. Although we may compare modelled coated-sphere thicknesses and compositions (Fig. 3) with published values for oceanic phytoplankton, a similar comparison would not be possible with detrital matter and particle aggregates. When considering only phytoplankton, the modelled high refractive coats of the spheres could represent cell walls made of calcite or cellulose and other polysaccharides[50]. However, coats with high refractive indices could also represent cellular and plastidial membranes composed by a bilayer of lipids with embedded proteins[50]. Moreover, the relative volume of the coat parameterised for coated spheres is similar to the relative cellular volume that chloroplasts and thylakoids generally occupy[41], whereas the cell wall contributes <2% of the cellular volume[41]. Previous studies have suggested that cellular organelles may have a significant influence on $b_{bp}$[28]. Highly refractive cell walls enhance the oceanic backscattering[61]. The outer shell of a coated sphere therefore most likely combines the effects that both external and internal cellular structures have on $b_{bp}$, though what is the main driver of $b_{bp}$ still remains unclear. Finally, how the modelled coated spheres relate to oceanic detrital particles remains to be understood.

There is still much to do to mechanistically understand open-ocean particle backscattering observations. We need to characterise better the internal and external structures of marine organisms (e.g., cell wall and membrane thickness; number, size and position of intracellular organelles and starch inclusions) and how they vary with depth and latitude. We also need to characterise better the detrital matter and particle aggregates. For example, it is possible that detritus constitutes a large fraction of oceanic particles[62] but its abundance, structural complexity and distribution along the full size range are still largely unknown. To characterise particle structural characteristics, integrated approaches based on microscopic and imaging analysis, coupled with optical measurements in situ and in the laboratory, such as polarised light scattering[36,63], are needed. For example, by comparing simulated (see Methods) and measured[64] polarised components of the Mueller scattering matrix[65], the range of selected coated-sphere parameterisations may be further constrained (Supplementary Fig. 10). The comparison of our simulated polarised elements with in-situ measurements for a range of oceanic waters[64] suggests indeed that oceanic particles may be better represented by high refractive indexes and low thicknesses of the outer shells, and further shows the inadequacy of the homogeneous-sphere assumption (Supplementary Fig. 10). Characterising the complexity of marine particles in situ is thus the necessary step towards identifying the main drivers of the optical backscattering. This step will also tell us if a coated sphere is indeed sufficient to explain the sources of the open-ocean backscattering.

Our findings clearly show that the structural complexity of particles is key to explain mechanistically the sources of open-ocean backscattering. In addition to measuring the enigmatic open-ocean submicron particles[32], future efforts should also be directed to test and constrain complex optical models with in-situ measurements of particle and cellular characteristics. Only when we will understand the sources of oceanic $b_{bp}$, we will be able to exploit the full potential of optical backscattering observations for investigating the biological carbon pump. Our study opens a new direction towards achieving this ultimate goal.

## Methods

**Cruise location and sampling strategy**. Measurements were collected during two Atlantic Meridional Transect (AMT) cruises carried out in October–November 2012 (AMT22) and in September–November 2016 (AMT26). Both cruises covered the Atlantic Ocean from the United Kingdom to South America, by encompassing a wide range of oceanic conditions, from sub-polar to tropical and from eutrophic systems to mid-ocean oligotrophic gyres (Supplementary Fig. 1). During both cruises, we collected measurements of PSD (i.e., the number of particles per unit volume per unit of particle size; units of particles $\text{m}^{-3}\,\mu\text{m}^{-1}$), and particulate beam attenuation and optical

backscattering coefficients at the wavelength 532 nm ($c_p(532)$ and $b_{bp}(532)$, respectively). Here, we use the particulate beam attenuation coefficient as a measure of the total scattering because these two quantities are numerically close at 532 nm due to negligible particulate light absorption[25].

During AMT22, PSD and optical measurements were collected at 134 stations in flow-through mode from the 5-m underway clean seawater supply of the ship (Supplementary Fig. 1). On AMT26, a rosette system equipped with $24 \times 20$-l Niskin bottles and sensors for hydrological and optical measurements was deployed in the upper 500 m at 23 stations (Supplementary Fig. 1). Up-cast seawater collection for PSD analysis was performed at five depths (5, 150/200, 300, 400/450, and 500 m) in addition to the DCM as detected by an AquaTracka III fluorometer (Chelsea Technologies Group Ltd). Temperature and salinity measurements were acquired during the cast by a Sea-Bird Scientific SBE 9 Conductivity-Temperature-Depth (CTD) sensor.

A size-fractionation experiment was conducted during the AMT26 cruise to assess the contribution of submicron particles (i.e., <1 μm) to optical scattering measurements. A total of 22 additional stations were sampled, at the same time as the ship's CTD casts, in flow-through mode from the 5-m underway ship's seawater supply (Supplementary Fig. 1). Optical measurements were taken, and seawater for PSD analysis collected, before and after filtration through a pre-rinsed (with ultra-pure water) large surface-area cartridge filter (Cole Parmer) with nominal pore size of 1 μm (i.e., AMT26_1μm). Filter back-flushing using ultra-pure water before filtrations minimised filter clogging. Filtering in flow-through mode minimised particle aggregation and precipitation. During filtration, the flow rate of the underway system was set to 2 l min$^{-1}$.

AMT26 data collected at the ocean surface (5 m) and at the DCM were used to test the homogeneous-sphere assumption for marine particles and parameterise the coated-sphere model. AMT22 and AMT26_1μm samples were used as independent datasets to validate model results. AMT26 samples collected in the mesopelagic region of the Atlantic Ocean were used to discuss coated-sphere model advantages and limitations.

**Particle size distribution (PSD).** During AMT22, a total of 134 PSDs were measured on board, in multiple 2-ml replicates (from 2 to 26), through a Multisizer III Coulter counter (Beckman Coulter) fitted with a 70-μm aperture. The measured size range of equivalent spherical diameters (ESDs) was 1.4–42 μm and distributed in 200 logarithmically spaced size bins. During AMT26, seawater samples ($N = 134$ from casts + 44 from underway) were directly collected in 500-ml acid-washed amber bottles and immediately prepared under a vertical flow hood for the analysis. Samples were kept in a cold and dark place during the analysis. Measurements were conducted with a Multisizer III Coulter counter (Beckman Coulter) fitted with 20-μm and 100-μm apertures, in multiple 50-μl and 1-ml replicates, respectively. The ESD size ranges were 0.588–12 μm and 2–60 μm for the 20-μm and 100-μm apertures, respectively, each distributed in 256 logarithmically spaced size bins. The number of replicates was selected, sample by sample, to achieve an overall error <15% in a given size range (~3 μm and 6 μm for 20-μm and 100-μm apertures, respectively), which corresponded to a detection limit of 45 counted particles (i.e., from three repetitions at the DCM up to 15 for samples in the mesopelagic zone).

For both AMT22 and AMT26 samples, the PSD was calculated for each aperture by summation of all the replicates. The total analysed volume was 4–52 ml with the 70-μm aperture, 150–600 μl with the 20-μm aperture, and 3–15 ml for the 100-μm aperture. The accuracy of measurements within each size bin was calculated through the standard law of propagation of uncertainty[66] considering the number of counted particles, the blank reference and the Multisizer III volumetric pump accuracy (0.5%) as sources of error. Specific to the AMT26, the PSDs obtained from the 20-μm and 100-μm apertures were combined in a single PSD by merging the measurements from the two apertures at around 2.14 μm, where they presented similar bin width and upper/lower limits. The resulting PSD spanned from 0.588 to 60 μm by 360 logarithmically spaced size bins with associated uncertainties.

Before each cruise, the Coulter counter apertures were calibrated using suspensions of recommended Beckman Coulter calibration spheres. During AMT26, the instrument performance was checked using suspensions of standardised spheres of 3.6 μm of diameter (Beckman Coulter) and provided counting accuracy within 10% for both 20-μm and 100-μm apertures. Blank references of 500-m seawater filtered through 0.1-μm Millex sterile polyvinylidene difluoride syringe filters (Millipore) were recorded daily for both 20-μm and 100-μm apertures. During AMT22, blanks were acquired through repeated (three times) filtration of surface seawater on 0.2-μm syringe filters.

Examples of PSDs from AMT22, AMT26, and AMT26_1μm with combined uncertainties are shown in Supplementary Figure 11. Supplementary Figure 12 shows examples of PSDs simultaneously collected during AMT26 from the ship's underway and using Niskin bottles at 5 m depth. No significant differences in PSDs were observed between the two sampling platforms.

**In-situ particulate optical scattering.** Bulk beam attenuation coefficients at 532 nm ($c(532)$) were measured with an ac-s spectrophotometer (WETLabs, Seabird-Scientific) for underway samples (AMT22 and AMT26_1μm), or through 0–250 m casts of an ac-9 spectrophotometer (WETLabs, Seabird-Scientific) at the same time

of CTD measurements and water sampling. The ac-s and ac-9 had the same optical configuration and beam acceptance angle[67], and collected data along a 25-cm path length. Both underway and cast measurements were repeated after filtration through a 0.2-μm filter in order to determine the contribution of the coloured dissolved organic matter, as well as calibration drifts[31]. The beam attenuation coefficient of particles ($c_p(532)$) was obtained by subtraction of the 0.2-μm filtered signal from $c(532)$, following established protocols[52].

The angular scattering coefficient at 532 nm was measured at a central angle of 124°, $\beta(124, 532)$, with three ECO-BB3 backscattering sensors (WETLabs, Seabird-Scientific). During AMT26, the ECO-BB3 was installed on the rosette sampling system and provided 0–500 m $\beta(124, 532)$ measurements. For AMT22 and AMT26_1μm underway samples, $\beta(124, 532)$ coefficients were acquired in a flow-through chamber. Data were collected and processed following established protocols[31,52]. The relative uncertainty associated with the extrapolation of the optical backscattering coefficient from a single angle measurement is <10%[68,69].

All $c_p(532)$ and $b_{bp}(532)$ measurements were quality checked[31,52]. Underway measurements were binned into 1-min intervals and the median value was calculated for the timing corresponding to PSD water sampling. The $c_p(532)$ and $b_{bp}(532)$ coefficients acquired during upward casts were binned in 1-m intervals and the median was calculated for values within a 5-m window centred at the depth of the Niskin bottles fired for the PSD samples. Before 1 m binning, $b_{bp}(532)$ profiles were smoothed with a moving-median filter (five-point window). To assess the robustness of the optical measurements used in the analysis, we checked consistency through inter-comparisons among them and versus independent simultaneous measurements when available (e.g., HobiLabs HydroScat-6P; Supplementary Figs. 13 and 14).

**Modelling particulate optical scattering properties.** The beam attenuation and optical backscattering coefficients of particles were modelled at the wavelength 532 nm as follows:[27]

$$c_p(n) = \frac{\pi}{4} \int_{D_{min}}^{D_{max}} Q_c(D, n) D^2 F(D) \mathrm{d}D \tag{1}$$

and

$$b_{bp}(n) = \frac{\pi}{4} \int_{D_{min}}^{D_{max}} Q_{b_b}(D, n) D^2 F(D) \mathrm{d}D \tag{2}$$

where $D$ is the particle diameter (units of m) with minima and maxima as defined from PSD measurements (i.e., $(0.588-60) \times 10^{-6}$ m for AMT26 and $(1.4-42) \times 10^{-6}$ m for AMT22 samples). We did not extrapolate the PSDs for particles with diameters $<D_{min}$ that consequently were not modelled. For each individual particle, $Q_c(D, n)$ and $Q_{b_b}(D, n)$ are the dimensionless efficiency factors for beam attenuation and optical backscattering respectively, at 532 nm, and $n$ is the volume-averaged refractive index[38], which represents the characteristics of the coated sphere with refractive indices (relative to that of seawater[54]) of its core and coat, $n_1$ and $n_2$, respectively, and fractional thickness of the coat $tk_2$; the term $\pi D^2/4$ indicates the particle cross-sectional area (units of m$^2$); and $F(D)$ is the concentration of particles for a given diameter as obtained from the PSD measurements (units of particles m$^{-4}$). The values of $n_2$ were varied between 1.02 and 1.22[50], whereas those of $tk_2$ varied between 2 and 30% of the radius of the particle. The volume-averaged refractive index of the particle was forced to be equal to 1.06, which was the single value that best fitted the beam attenuation measurements through the homogeneous-sphere assumption (see Results). As a consequence, the refractive index of the core $n_1$ varied as a function of $n_2$ and $tk_2$. A total of 609 combinations of $tk_2$ and $n_2$ were tested with the coated-sphere model. Homogeneous spheres were represented as a coated sphere with $tk_2 = 0$ and $n_1$ values between 1.02 and 1.22[50]. Particles counted within each size bin of the measured PSD were considered as monodispersed and non-absorbing populations at the selected wavelength[25]. We expect the results of this modelling analysis to be more sensitive to the structural complexity of the spheres than to the impact of light absorbing material on $b_{bp}$[38,39], and thus to be representative also of other portions of the visible spectrum.

Look-Up-Tables of $Q_c(D, n)$ for homogeneous and coated spheres were generated by using the scattnlay software[70], for angles between 0 and 180° (0.1° increments) and logarithmically spaced ESDs between 0.4 and 70 μm. The $Q_c(D, n)$ values both for homogeneous and coated spheres were corrected for the acceptance angle effect to be consistent with WETLabs ac-9 and ac-s in-situ measurements[67].

The values of $Q_{b_b}(D, n)$ were computed as:

$$Q_{b_b}(D, n) = Q_b(D, n) \frac{\int_{\pi/2}^{\pi} S_{11}(\theta, D, n) \sin\theta \mathrm{d}\theta}{\int_0^{\pi} S_{11}(\theta, D, n) \sin\theta \mathrm{d}\theta} \tag{3}$$

where $Q_b$ is the dimensionless efficiency factor for the total scattering (output of the freely available scattnlay software[70]) and $S_{11}(\theta, D, n)$ is the top left component (i.e., 11) of the scattering matrix computed as described below. To compute the VSF and the other non-normalised components of the Mueller matrix we proceeded as follows[70]. First, for each set of particle parameters (i.e., $D$ and $n$ for the wavelength 532 nm) we computed the following elements of the scattering

matrix:

$$S_{11}(\theta, D, n) = 0.5\left[|S_2(\theta, D, n)|^2 + |S_1(\theta, D, n)|^2\right] \quad (4)$$

$$S_{12}(\theta, D, n) = 0.5\left[|S_2(\theta, D, n)|^2 - |S_1(\theta, D, n)|^2\right] \quad (5)$$

$$S_{33}(\theta, D, n) = 0.5\left[S_1(\theta, D, n)S_2^*(\theta, D, n) + S_1^*(\theta, D, n)S_2(\theta, D, n)\right] \quad (6)$$

$$S_{34}(\theta, D, n) = 0.5i\left[S_1(\theta, D, n)S_2^*(\theta, D, n) - S_1^*(\theta, D, n)S_2(\theta, D, n)\right] \quad (7)$$

where $S_1(\theta, D, n)$ and $S_2(\theta, D, n)$ are the complex scattering amplitude outputs of the scattnlay software[70] and the starred quantities indicate complex conjugates.

Each $S_{ij}(\theta, D, n)$ component was then normalised by the integral of the $S_{11}(\theta, D, n)$ component to compute normalised elements of the scattering matrix:

$$\overline{S_{ij}}(\theta, D, n) = \frac{S_{ij}(\theta, D, n)}{2\pi \int_0^\pi S_{11}(\theta, D, n)\sin\theta\,\mathrm{d}\theta} \quad (8)$$

We then computed the Mueller matrix components of particle populations composed of modelled coated spheres and the measured PSDs, $F(D)$, as:

$$M_{ij}(\theta, n) = \frac{\pi}{4}\int_{D_{\min}}^{D_{\max}} \overline{S_{ij}}(\theta, D, n)Q_b(D, n)D^2 F(D)\,\mathrm{d}D \quad (9)$$

with $M_{11}(\theta, n)$ resulting in the modelled VSFs.

To compare the shape of the VSFs with existing measurements, we normalised $M_{11}(\theta, n)$ by its value integrated between 0 and $\pi$ (i.e., total scattering) to obtain the corresponding phase functions as:

$$\text{Normalized VSF}(\theta, n) = \frac{M_{11}(\theta, n)}{2\pi \int_0^\pi M_{11}(\theta, n)\sin\theta\,\mathrm{d}\theta} \quad (10)$$

Finally, to compare the polarised components with existing datasets, we normalised each component of the scattering matrix by $M_{11}(\theta, n)$ as follows:

$$P_{ij}(\theta, n) = \frac{M_{ij}(\theta, n)}{M_{11}(\theta, n)} \quad (11)$$

where now $i$ and $j$ are not simultaneously equal to 1.

**Statistics**. The combined uncertainty of modelled $c_p(532)$ and $b_{bp}(532)$ coefficients was calculated by propagation of the errors associated with PSD measurements[66]. Modelled $c_p(532)$ and $b_{bp}(532)$ coefficients were then compared with in-situ measurements through match-up analysis. The model performance was evaluated by calculating the systematic error, in logarithmic space, as follows:

$$\text{Bias} = \text{median}\left(\log_{10}(\bar{x}_i) - \log_{10}(x_i)\right) \quad (12)$$

where $\bar{x}_i$ and $x_i$ are the modelled and measured values, respectively. The Pearson's correlation coefficient ($r$) was also calculated between log-transformed quantities and the significance was verified by a two-tailed Student's $t$-test (confidence level equal to 99%; $N$-2 degrees of freedom).

The cumulative percent contributions to $c_p(532)$ and $b_{bp}(532)$ of the size fraction from $D_{\min}$ to $D$ were defined, respectively, as:[27]

$$C_{c_p}(D) = 100\int_{D_{\min}}^{D} Q_c(D, n)D^2 F(D)\,\mathrm{d}D\left(\int_{D_{\min}}^{D_{\max}} Q_c(D, n)D^2 F(D)\,\mathrm{d}D\right)^{-1} \quad (13)$$

and

$$C_{b_{bp}}(D) = 100\int_{D_{\min}}^{D} Q_{b_b}(D, n)D^2 F(D)\,\mathrm{d}D\left(\int_{D_{\min}}^{D_{\max}} Q_{b_b}(D, n)D^2 F(D)\,\mathrm{d}D\right)^{-1} \quad (14)$$

### Data availability

This study uses data from the Atlantic Meridional Transect Consortium (NER/0/5/2001/00680), provided by the British Oceanographic Data Centre and supported by the Natural Environment Research Council. Particle size distribution data collected during the AMT26 cruise can be downloaded at https://doi.org/10.5285/79103bda-8913-39f3-e053-6c86abc0567a. All other data are freely available at https://www.bodc.ac.uk/projects/uk/amt/ and https://www.bodc.ac.uk/data/bodc_database/nodb/data_collection/207/, or from the authors upon request.

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

## Acknowledgements

The authors would like to thank the captains, chief scientists, officers and crew of the RRS James Cook (UK National Oceanography Centre) and the RRS James Clark Ross (British Antarctic Survey) for the help they provided during AMT22 and AMT26 field expeditions. We thank Kristen Reifel and Virginie van Dongen-Vogels for helping in data collection during AMT22. We thank Hervé Claustre for insightful conversations. This study has been conducted in the frame of the REOPTIMIZE—REmineralisation, OPTIcs and Marine partICle siZE project. This project has received funding from the European Union's Horizon 2020 research and innovation programme under the Marie Skłodowska-Curie grant agreement No 706781 (assigned to E. Organelli and the Plymouth Marine Laboratory). This study is a contribution to the international IMBeR project and was supported by the UK Natural Environment Research Council National Capability funding to Plymouth Marine Laboratory and the National Oceanography Centre, Southampton. This is contribution number 326 of the AMT programme. Optics data collection by G.D. and R.J.W.B. during AMT26 was supported by the European Space Agency (ESA) contract: Copernicus Sentinel Atlantic Meridional Transect Fiducial Reference Measurements Campaign (AMT4SentinelFRM; grant ESRIN/RFQ/3-14457/16/I-BG to Gavin Tilstone and the Plymouth Marine Laboratory). Data collection by G. D. during AMT22 was funded by NASA (grant NNX10AT70G to Mike Behrenfeld and the Oregon State University). E.B. was supported by NASA (Grant NNX15AC08G). We thank Steven Lohrenz, Kenneth Voss, and an anonymous reviewer for their constructive comments on a previous version of the manuscript.

## Author contributions

E.O. and G.D. conceived and designed the study. E.O. collected and processed PSDs during AMT26, quality-controlled PSDs from AMT22 and optics data from both cruises, executed all the analysis, organised and wrote the manuscript, and prepared all figures and tables including those in the supplementary material. G.D. collected AMT22 PSDs, optics and phytoplankton pigments, and generated the look-up-tables of size-resolved scattering efficiency coefficients. G.D. and R.J.W.B. collected AMT26 optical measurements. R.J.W.B. organised the 1 µm filtration experiment and collected AMT26 phytoplankton pigments. G.A.T. performed flow cytometry analysis and assisted in measuring PSDs and in maintaining the Coulter Counter during AMT26. E.B. and A.B. provided insightful comments to early and advanced stages of the study. All authors commented on the final version of the manuscript.

## Additional information

**Competing interests:** The authors declare no competing interests.

