## [Peer Review File · Nature Communications]

Reviewers' comments:

Reviewer #1 (Remarks to the Author):

This is an important contribution examining the roles of different particle size classes in contributions to optical backscattering in oceanic waters. Understanding the factors that influence backscattering is critical to being able to model and describe light propagation and interpret optical signatures detected by in situ and above water sensors (e.g., satellite ocean color).

The authors apply a coated-sphere model to simulate the characteristics of particulate beam attenuation and backscattering for a large surface ocean data set collected as part of the Atlantic Meridional Transect cruises. They also conducted particle size distributions and size fractionation experiments, providing insight as to how different particle size classes contribute to scattering. The methodology appears to be robust and the authors provide comparisons for underway and Niskin bottle samples as well as for different methods of estimating backscattering.

The authors examined a range of possible combinations of refractive index and thickness of the outer coat, and were able to identify a subset of realistic combinations. They also considered observations for the mesopelagic and noted that particles in this depth range appeared to have different characteristics than was simulated by the average coated-sphere model, and thus would require a different parameterization.

The study focuses exclusively on results for the 532 nm wavelength, which is typically representative of maximum light penetration in the water column and accordingly a relatively high water-leaving radiance in open ocean waters. It would be useful for the authors to make some mention of how representative (or not) these findings would be for other wavelengths in the visible spectrum, given the differing contributions of absorption and scattering the different relationships of particle size and backscattering for other wavelengths.

Overall, this manuscript presents a novel explanation for the "missing backscatter" at least for the 532 nm wavelength observations. Clearly, the coated-sphere model is still an oversimplification of the complex and diverse range of particles in the ocean, but the authors have made a compelling case that it does a better job than the traditional homogeneous sphere approach, and they provide some explanation for why it may be more representative of various types of particles and aggregations of particles. This work has the potential to have a significant impact on interpretations of scattering in natural waters, and may help to improve our ability to infer particle characteristics from optical signatures. There are some relatively minor revisions that could be made to improve the manuscript and provide more clarity in some instances. Provided these are addressed, I recommend publication.

Specific Comments:

- 1) The authors should make a mention early on in the manuscript that they are focusing on the 532 nm wavelength and explain why this is justified and how representative it (or not) of other portions of the spectrum.
- 2) P. 3, Lines 76-77 – Add a comment that contributions to scattering by different size fractions will be examined.
- 3) P. 3, Line 101 – It would be helpful to add a statement that although the coated-sphere model is still a simplification, some explanation of why this may be more representative of various types of oceanic particles will be provided.
- 4) P. 6, Lines 168-169 – Be explicit that this result was based on results with the coated-sphere model.
- 5) P. 7, Line 179 – Change "particle diameters" to "size threshold of particle diameters" or similar.
- 6) P. 7, Line 188 – A value of 40% is still a substantial percentage. Also, filtration may remove particles smaller than the nominal pore size (as a result of clogging, aggregation, etc.). Filtration may also alter particle characteristics. The authors should acknowledge some of these points.
- 7) P. 8, Line 219 - I question the use of "accurately" here. I would agree that the coated sphere

model did a better job of characterizing the attenuation and backscattering, but not without some remaining bias. Perhaps restate as “with reduced bias” or similar.

8) P. 15, Lines 468-469 – A brief explanation of how total particulate backscattering is extrapolated on the basis of a single angle measurement would be useful. Also, the authors should comment on the degree to which the extrapolation based on a single angle may introduce additional uncertainty in their analysis.

Steven Lohrenz

Reviewer #2 (Remarks to the Author):

The authors present a study comparing measurements of particulate attenuation and backscattering collected across multi-province oceanic scales with modelled estimates of the same characteristics based on Mie theory and a series of parameterised versions of a ‘coated-sphere’ model. The authors produce a seemingly convincing demonstration that the former is incapable of reproducing the observations (specifically simultaneously producing low bias in both backscattering and attenuation), in marked contrast to the result when using a range of realistic parameterisations of the latter. As outlined in a number of places by the authors, the work has important implications for our understanding of both the relationship between the optical properties of oceanic waters and the structural complexities of the particles they contain and hence our broader interpretation of ocean optical measurements derived both in situ and remotely. Overall I found the manuscript to be well written and generally clear. The analysis and arguments build on extensive prior work on the coated-sphere model but none-the-less the work appears appear novel and well supported and I would expect the work to be of broad interest to the field. Below I outline a few remaining, albeit relatively minor, comments the authors might like to address.

General comments

Although effectively covered in the discussion (Lines 348-374), for the more general reader it might be worth briefly introducing why ‘heterogeneous coated spheres’ might provide reasonable approximations for the characteristics of a range of marine particles further towards the beginning of the manuscript (e.g. Line 65).

Given the style of manuscript places the methods at the end, I wondered whether some introductory sentences summarising what was done might help general readability at the beginning of the results section.

Line 105 and subsequent use: personally I found the use of the term ‘average model’ to describe the presented model results a bit confusing on first encounter and wondered if ‘model ensemble’ might be a more representative/intuitive term for what the presented range of modelled results across the realistic parameter set (Figure 3). This could then be more precisely related to the ‘ensemble average’ (to replace ‘average model’) and ‘ensemble standard deviation’ within subsequent descriptions and figures as appropriate.

Lines 291-293: In addition to mixed populations of homogenous and coated spheres equally distributed across the resolved size range (Suppl. Fig. 6), was it possible to consider how robust the conclusions might be to the (presumably) more realistic cases where the relative contributions of such types of particles might vary across the size range and/or where there will actually be multiple populations of different types of coated-sphere, with characteristics (and hence parameters) again varying across the size range?

Additional minor comments:

Line 54: suggest '...in interpreting bbp arise due to...'

Line 348: there could be argued to be some degree of circularity in this sentence as stated given that parameter combinations which were not 'reasonable' had already been discarded (Figure 3).

Figure 4(b), you could consider including some indication of the range of bbp measurements within the training data on this plot.

Line 488: , suggest '... Q_c and Q_{bb} are the dimensionless efficiency factors for beam attenuation and backscattering respectively...'

Reviewer #3 (Remarks to the Author):

Review Organelli et al. The open ocean missing backscattering....

This paper points out that by using a two layer sphere model the same particles can match both c_p and associated bbp measurements. This is important because usually it is thought that the two parameters (c_p and bbp) are driven by different size classes of particles, large and small, and there has been a problem with not finding enough small particles to backscatter to match measurements in the ocean. Layered sphere calculations are not new, people have been doing them at least since Bohren and Hoffman published the computer code to calculate this in their book in the early 80's. And people have used layered spheres to match specific cases, or to explain volume phase functions or polarization matrix elements, but I don't think I have seen anyone try to find somewhat "global" or at least basin wide parameters for the spheres that, when using measured PSD's, match both c_p and bbp . And the fact that you can is the important point of this paper.

The paper is well written and clear, particularly the first 6 pages when the main points of the paper are discussed. I have some specific comments listed below. A couple of things that are missing, most important would be an example of the VSF that is calculated with one of their PSD's and sphere model (or a lot of them...is there a regular shape to them, perhaps plot the phase function instead?). Is it realistic looking (besides the integrated value of bbp). Personally, I would like to see a few of the Mueller matrix elements, but while they say they calculated the VSF to get bbp from their software, the Mueller matrix elements would take more calculation, and the interest is not wide spread and not a point of this paper.

I do have an issue with this, and most of the other papers dealing with Mie or more complicated sphere models of particles, and that is giving the parameters (such as thickness of layer or index of refraction) to any real physical parameter of the phytoplankton. The derived thickness or index of refraction is the number needed to make this models c_p and bbp "equivalent" to the populations values for c_p and bbp . It is great that the same model can be used to calculate both of these parameters, but I don't think it says a lot about the particles themselves. As the authors say, some particles maybe sufficiently modeled as homogeneous spheres, but very few useful particles are really homogeneous spheres. My concern here is towards the end of the paper where the discussion gets more speculative and wide ranging, that it goes too far at times.

Specific comments

Line 195: Not sure why the sentence starts with "in contrast to what Mie theory predicts" Mie

theory predicts what it does for homogeneous spheres, coated spheres do something else....Sentence would make more sense without this phrase.

Line 205, figure 6a, I don't understand this figure at all. It is the ratio of what?
Figure 6b, I think 40 % of the backscattering being less than 1um is a fairly significant amount. Did you consider this when choosing parameters for the layered sphere (that you only had to come up with 60% of the backscattering?). Too bad the size fractionation experiment was not done more often, or that you had more data about the <1um component.

Lines 258 to 271: this paragraph doesn't really add anything to the paper or your case and is fairly obvious.

Lines 285 -288. You first say that if particles are homogeneous spheres they will backscatter less...fine, that is the point of the paper. But then the next sentence says that this is because submicron particles backscatter more...these two sentences independently are fine (particularly if you take out "This is because"), but the reasoning doesn't really flow at all..

Line 310-311. This statement seems to be very repetitive in the article. By now the reader should understand (and has read several times in the paper) that backscattering is sensitive to particle structural and compositional characteristics.

Line 342-344. Are you trying to say that cp and bbp are "grossly" correlated with each other across large regions, but not in the fine scale (hence anti correlation in diel cycles)? The two statements, correlated and exhibiting different behaviors, seem conflicted.

Line 369; coincidentally, a paper just came out in Applied Optics about coated spheres and why they are needed to explain polarization in the ocean.

Line 495: there is a step missing here. The way it sounds Scatnlay gives you VSF's and then you have to integrate them to get Qbb.

Kenneth Voss

Revision of the manuscript

The open-ocean missing backscattering is in the structural complexity of particles

by Organelli et al.

We sincerely thank Dr. Steven Lohrenz, Dr. Kenneth Voss and the anonymous Reviewer for the time they invested in reviewing our manuscript and for providing constructive comments and suggestions. Hereafter, the reviewer's comments appear in black, while our responses (**R#**) and actions (**A#**) are in blue. All modifications are highlighted in red in the tracked-changes version of the manuscript.

Reviewers' comments:

Reviewer #1 - Steven Lohrenz:

This is an important contribution examining the roles of different particle size classes in contributions to optical backscattering in oceanic waters. Understanding the factors that influence backscattering is critical to being able to model and describe light propagation and interpret optical signatures detected by in situ and above water sensors (e.g., satellite ocean color).

The authors apply a coated-sphere model to simulate the characteristics of particulate beam attenuation and backscattering for a large surface ocean data set collected as part of the Atlantic Meridional Transect cruises. They also conducted particle size distributions and size fractionation experiments, providing insight as to how different particle size classes contribute to scattering. The methodology appears to be robust and the authors provide comparisons for underway and Niskin bottle samples as well as for different methods of estimating backscattering.

The authors examined a range of possible combinations of refractive index and thickness of the outer coat, and were able to identify a subset of realistic combinations. They also considered observations for the mesopelagic and noted that particles in this depth range appeared to have different characteristics than was simulated by the average coated-sphere model, and thus would require a different parameterization.

The study focuses exclusively on results for the 532 nm wavelength, which is typically representative of maximum light penetration in the water column and accordingly a relatively high water-leaving radiance in open ocean waters. It would be useful for the authors to make some mention of how representative (or not) these findings would be for other wavelengths in the visible spectrum, given the differing contributions of absorption and scattering the different relationships of particle size and backscattering for other wavelengths.

R1: While we agree with the Reviewer that optical backscattering can decrease within the absorption bands of algal pigments and detrital matter, we believe this is a second-order problem and thus our findings for the 532 nm wavelength will apply, to first order, to other wavelengths more affected by absorption. We now provide more details to support this statement. Early studies (e.g., Bricaud et al., 1992; Zaneveld and Kitchen, 1995) investigated the spectral variations of the optical backscattering efficiency (i.e., Q_{bb}) and coefficient (i.e., b_b) in phytoplankton by assuming both multi-layered and homogeneous spherical geometries for particles. These studies independently observed that, although spectral variations in Q_{bb} and b_b exist, the values derived from coated-sphere models are up to orders of magnitude higher than those obtained for the equivalent (in size and average composition) homogeneous sphere along the full spectrum of the visible light (see Fig. 2c in Bricaud et al. 1992 and Fig. 9a in Zaneveld and Kitchen 1995). These findings have been later confirmed for various phytoplankton species by Bernard et al. (2009, Biogeoscience discussion, www.biogeosciences-discuss.net/6/1497/2009/) in their Fig. 3d and Fig. 6d. Light absorption would therefore minimally impact our results and the resolution of the missing backscattering enigma.

A1: Lines 80–84 (revised ms): the following sentences have been added to explain why we have executed the analysis only at 532 nm “*We based our analysis on coincident measurements of particle size distributions (0.59-60 μ m) and optical measurements at the wavelength 532 nm (see methods). We performed the analysis only at 532 nm because it allows us to minimise the second-order effect of*

particulate light absorption on scattering (Morel and Bricaud, 1986), as well as to directly compare our results with previous results based on the homogeneous-sphere (Stramski and Kiefer, 1991)."

Lines 523-525 (revised ms): the following sentences have been added to explain that the results of this study made for the wavelength 532 nm can be representative also of other portions of the visible spectrum "*We expect the results of this modelling analysis to be more sensitive to the structural complexity of the spheres than to the impact of light absorbing material on b_{bp} (Bricaud et al., 1992; Zaneveld and Kitchen, 1995), and thus to be representative also of other portions of the visible spectrum.*"

Overall, this manuscript presents a novel explanation for the "missing backscatter" at least for the 532 nm wavelength observations. Clearly, the coated-sphere model is still an oversimplification of the complex and diverse range of particles in the ocean, but the authors have made a compelling case that it does a better job than the traditional homogeneous sphere approach, and they provide some explanation for why it may be more representative of various types of particles and aggregations of particles. This work has the potential to have a significant impact on interpretations of scattering in natural waters, and may help to improve our ability to infer particle characteristics from optical signatures. There are some relatively minor revisions that could be made to improve the manuscript and provide more clarity in some instances. Provided these are addressed, I recommend publication.

Specific Comments:

1) The authors should make a mention early on in the manuscript that they are focusing on the 532 nm wavelength and explain why this is justified and how representative it (or not) of other portions of the spectrum.

R2/A2: done. Please see **R1/A1** above.

2) P. 3, Lines 76-77 – Add a comment that contributions to scattering by different size fractions will be examined.

R3/A3: Lines 84-85 (revised ms): done.

3) P. 3, Line 101 – It would be helpful to add a statement that although the coated-sphere model is still a simplification, some explanation of why this may be more representative of various types of oceanic particles will be provided.

R4/A4: we have added this information in the introduction at **Lines 65-70** (revised ms) according to Reviewer#2 suggestion. Please also see **R10/A10** below.

4) P. 6, Lines 168-169 – Be explicit that this result was based on results with the coated-sphere model.

R5/A5: Line 175 (revised ms): done.

5) P. 7, Line 179 – Change "particle diameters" to "size threshold of particle diameters" or similar.

R6/A6: Captions of Figure 5 and revised Supplementary Figure 6: done.

6) P. 7, Line 188 – A value of 40% is still a substantial percentage. Also, filtration may remove particles smaller than the nominal pore size (as a result of clogging, aggregation, etc.). Filtration may also alter particle characteristics. The authors should acknowledge some of these points.

R7/A7: Line 189 (revised ms): we have replaced the term "*small*" with "*smaller-than-expected*" (i.e., with respect to the homogeneous-sphere model).

Lines 192-195 (revised ms): the following sentences have been added "*Tests on the efficiency of these filtrations indicated that the filter retained the majority of particles for diameters closest to and above its nominal pore size limit (i.e., 1 μm), while progressively increasing amounts of particles with diameters smaller than 1 μm passed through the filter (Fig. 6a).*"

Lines 429-430 (revised ms): in the method section, we now make the reader aware that we have minimized clogging and particle aggregation/precipitation during the size-fractionation experiment: "*Filter back-flushing using ultra-pure water before filtrations minimized filter clogging. Filtering in flow-through mode minimized particle aggregation and precipitation.*"

7) P. 8, Line 219 - I question the use of "accurately" here. I would agree that the coated sphere model did a better job of characterizing the attenuation and backscattering, but not without some remaining bias. Perhaps restate as "with reduced bias" or similar.

R8/A8: Lines 229-233 (revised ms): the sentence now reads "*Here we showed that, by modelling open-ocean particles as a population of coated spheres, we simultaneously predicted the measured beam attenuation and backscattering coefficients with considerably smaller biases than those obtained using the homogeneous-sphere model, over a wide range of trophic conditions across the sunlit Atlantic Ocean.*"

8) P. 15, Lines 468-469 – A brief explanation of how total particulate backscattering is extrapolated on the basis of a single angle measurement would be useful. Also, the authors should comment on the degree to which the extrapolation based on a single angle may introduce additional uncertainty in their analysis.

R9: The uncertainty associated to the extrapolation of the optical backscattering coefficient from the volume scattering function measured at a single angle has been described widely by published modelling and experimental studies (Boss and Pegau, Applied Optics 2001; Sullivan and Twardowski, Applied Optics, 2009; Dall’Olmo et al., Biogeosciences 2009; Sullivan et al., Light Scattering Reviews 2013). In particular, the conclusions of the experimental work by Sullivan and Twardowski (2009) state that "*under most oceanic conditions, estimates of the particulate backscattering coefficient, using single angle scattering measurements near 110° to 120° and suitable conversion factors, are justified and should have a maximum uncertainty of less than a few percent once instrument noise is accounted for.*"

A9: Lines 489-490 (revised ms): the following sentence has been added in the method section "*The relative uncertainty associated to the extrapolation of the optical backscattering coefficient from a single angle measurement is less than 10% (Boss and Pegau, 2001; Sullivan and Twardowski, 2009).*" The references Boss and Pegau (2001) and Sullivan and Twardowski (2009) have been added.

Reviewer #2:

The authors present a study comparing measurements of particulate attenuation and backscattering collected across multi-province oceanic scales with modelled estimates of the same characteristics based on Mie theory and a series of parameterised versions of a ‘coated-sphere’ model. The authors produce a seemingly convincing demonstration that the former is incapable of reproducing the observations (specifically simultaneously producing low bias in both backscattering and attenuation), in marked contrast to the result when using a range of realistic parameterisations of the latter. As outlined in a number of places by the authors, the work has important implications for our understanding of both the relationship between the optical properties of oceanic waters and the structural complexities of the particles they contain and hence our broader interpretation of ocean optical measurements derived both in situ and remotely. Overall I found the manuscript to be well written and generally clear. The analysis and arguments build on extensive prior work on the coated-sphere model but none-the-less the work appears appear novel and well supported and I would expect the work to be of broad interest to the field. Below I outline a few remaining, albeit relatively minor, comments the authors might like to address.

General comments

Although effectively covered in the discussion (Lines 348-374), for the more general reader it might be worth briefly introducing why ‘heterogeneous coated spheres’ might provide reasonable approximations for the characteristics of a range of marine particles further towards the beginning of the manuscript (e.g. Line 65).

R10/A10: Lines 65-70 (revised ms): the following sentences have been added "*Heterogeneous coated spheres represent the simplest way to model the bulk structural complexity of marine particle*

populations. Coated spheres can reasonably represent the external and internal cellular structures (e.g., cell wall and chloroplasts) of marine spherical and non-spherical phytoplankton, and account for differences in their chemical nature (Poulin et al., 2018). Coated spheres can also be used to simulate aggregates of living and detrital particles and algal colonies.” The reference Poulin et al. (Journal of Quantitative Spectroscopy & Radiative Transfer, 2018), recently published, has been added. The sentences in Lines 77-80 of the previous version of the manuscript have been removed from the revised version to avoid redundancy.

Given the style of manuscript places the methods at the end, I wondered whether some introductory sentences summarising what was done might help general readability at the beginning of the results section.

R11/A11: Lines 84-86 (revised ms): an additional sentence has been included at the end of the introduction where methods have been briefly summarised. The sentence reads: “*We then examined the contributions to modelled scattering of different size fractions and confirmed the findings by additional optical measurements collected through independent size-fractionation experiments.*”

Line 105 and subsequent use: personally I found the use of the term ‘average model’ to describe the presented model results a bit confusing on first encounter and wondered if ‘model ensemble’ might be a more representative/intuitive term for what the presented range of modelled results across the realistic parameter set (Figure 3). This could then be more precisely related to the ‘ensemble average’ (to replace ‘average model’) and ‘ensemble standard deviation’ within subsequent descriptions and figures as appropriate.

R12/A12: replaced throughout manuscript, figures and supplementary materials.

Lines 291-293: In addition to mixed populations of homogenous and coated spheres equally distributed across the resolved size range (Suppl. Fig. 6), was it possible to consider how robust the conclusions might be to the (presumably) more realistic cases where the relative contributions of such types of particles might vary across the size range and/or where there will actually be multiple populations of different types of coated-sphere, with characteristics (and hence parameters) again varying across the size range?

R13: The reviewer raises a good point but to be able to answer this question we should have information describing, for example, the size distribution and structural complexity of detrital particles. This detailed information is unfortunately not available. Any assumption we could make to answer Reviewer’s question will produce high variability in the results (e.g., see Table 1 in Stramski et al., Applied Optics 2001) that we could not interpret in a reliable manner. We acknowledge that the example shown in Supplementary Figure 6 (Supplementary Figure 7 in the revised manuscript) is based on the simplest assumption that the fraction of homogeneous particles is equally distributed along the whole size range. Nonetheless, this assumption is the most intuitive and may allow the results to be easily interpreted also by readers working in other research fields.

A13: Lines 379-381 (revised ms): we now specify that complementary measurements are needed to understand the structural complexity of marine particles along the full size range: “*For example, it is possible that detritus constitutes a large fraction of oceanic particles (Grob et al., 2007) but its abundance, structural complexity and distribution along the full size range are still largely unknown*”.

Additional minor comments:

Line 54: suggest ‘...in interpreting bbp arise due to...’

R14/A14: Line 55 (revised ms): done.

Line 348: there could be argued to be some degree of circularity in this sentence as stated given that parameter combinations which were not ‘reasonable’ had already been discarded (Figure 3).

R15: The reviewer raises a good point but to define the coated-sphere model we tested 609 combinations that did not all represent oceanic organic particles and/or their structures (see Supplementary Figure 4b). The lowest biases were independently produced for a set of combinations that also included those reasonable values for oceanic organic particles. However, there were other

combinations that could be representative of marine particles that yielded higher biases. We therefore used the information from the literature only to refine the model outputs among those combinations yielding the lowest biases.

A15: Lines 358-360 (revised ms): the sentence now reads “*The coated-sphere model predictions compared favourably to in-situ observations only when a specific set of parameters was selected. These selected parameters represent the known metabolite composition of oceanic particles (Aas, 1996).*”

Figure 4(b), you could consider including some indication of the range of bbp measurements within the training data on this plot.

R16/A16: done in Figure 4 as well as in revised Supplementary Figure 8. Captions have been modified accordingly.

Line 488: suggest ‘... Q_c and Q_{bb} are the dimensionless efficiency factors for beam attenuation and backscattering respectively...’

R17/A17: Lines 509-510 (revised ms): done.

Reviewer #3 – Kenneth Voss:

This paper points out that by using a two-layer sphere model the same particles can match both cp and associated bbp measurements. This is important because usually it is thought that the two parameters (cp and bbp) are driven by different size classes of particles, large and small, and there has been a problem with not finding enough small particles to backscatter to match measurements in the ocean. Layered sphere calculations are not new, people have been doing them at least since Bohren and Hoffman published the computer code to calculate this in their book in the early 80’s. And people have used layered spheres to match specific cases, or to explain volume phase functions or polarization matrix elements, but I don’t think I have seen anyone try to find somewhat “global” or at least basin wide parameters for the spheres that, when using measured PSD’s, match both cp and bbp. And the fact that you can is the important point of this paper.

The paper is well written and clear, particularly the first 6 pages when the main points of the paper are discussed. I have some specific comments listed below. A couple of things that are missing, most important would be an example of the VSF that is calculated with one of their PSD’s and sphere model (or a lot of them...is there a regular shape to them, perhaps plot the phase function instead?). Is it realistic looking (besides the integrated value of bbp). Personally, I would like to see a few of the Mueller matrix elements, but while they say they calculated the VSF to get bbp from their software, the Mueller matrix elements would take more calculation, and the interest is not wide spread and not a point of this paper.

R18: We acknowledge the lack of this important information in the previous version of the manuscript. We now show, in the Supplementary Materials, examples of both phase functions and P12/P33/P34 elements of the Mueller matrix as derived from measured particle size distributions and coated-sphere parameterisations. Both calculations have further confirmed the validity of our results, and thus the need of a coated-sphere model to represent the optical properties of oceanic particles. New Supplementary Figures (5 and 10) are shown, and the changes in the manuscript are detailed here below.

A18: Lines 120-121 (revised ms): we have added the following sentence “*The shapes of the volume scattering functions we obtained from these selected parameterisations were similar to existing measurements (Supplementary Fig. 5).*”

Supplementary Figure 5. Phase functions for the wavelength 532 nm (Normalized VSF; green solid lines) derived from particle size distributions (0.59–60 μm) measured at 5 m and at the DCM during the AMT26 cruise, by applying the coated-sphere model parameterisation with volume-averaged refractive index equal to 1.06, coat fractional thickness (tk_2) and refractive index (n_2) equal to 5% and 1.18, respectively (see Eq. 10 in the main text). The average of all VSFs is shown (black solid line). The average of VSFs as obtained from the same particle size distributions and the homogeneous-sphere model (refractive index equal to 1.06) is shown for comparison (blue dashed line). The VSF measured by Petzold (1972) at station 7 (09 July 1971), the “Tongue of the Ocean”, Bahama Islands, is shown as an example of in-situ clear waters (pink dotted line). The shapes of VSFs derived from the coated-sphere model are smooth and vary consistently with the measurement by Petzold (1972) as a function of the scattering angle. When using the homogeneous-sphere model, the shape of VSF shows a peak around 70° and severely underestimated backward values (angles >90°).

Lines 384-390 (revised ms): we have added the following sentences “*For example, by comparing simulated (see methods) and measured (Voss and Fry, 1984) polarized components of the Mueller scattering matrix (Bohren and Huffman, 1983), the range of selected coated-sphere parameterisations may be further constrained (Supplementary Fig. 10). The comparison of our simulated polarized elements with in-situ measurements for a range of oceanic waters (Voss and Fry, 1984) suggests indeed that oceanic particles may be better represented by high refractive indexes and low thicknesses of the outer shells, and further shows the inadequacy of the homogeneous-sphere assumption (Supplementary Fig. 10). The references Voss and Fry (Applied Optics, 1984) and Bohren and Huffman (1983) have been added.*

Supplementary Figure 10. Polarization components of the Mueller matrix as derived from the particle size distributions (0.59–60 μm) measured at 5 m and at the level of deep chlorophyll maximum during the AMT26 cruise (see Eq. 11 in the main text). The black solid lines represent the 12 selected coated-sphere parameterisations (see also Figure 3 in the main text) as derived from the average of all samples. The blue dashed line is obtained by assuming particles as homogeneous sphere and refractive index equal to 1.06. Red open circles and shaded areas are the average values and standard deviations, respectively, for a range of oceanic waters as reported in Tables 3 (for P34) and 6 (for P12 and P33) by Voss and Fry (1984).

I do have an issue with this, and most of the other papers dealing with Mie or more complicated sphere models of particles, and that is giving the parameters (such as thickness of layer or index of refraction) to any real physical parameter of the phytoplankton. The derived thickness or index of

refraction is the number needed to make this models c_p and b_{bp} “equivalent” to the populations values for c_p and b_{bp} . It is great that the same model can be used to calculate both of these parameters, but I don’t think it says a lot about the particles themselves. As the authors say, some particles maybe sufficiently modeled as homogeneous spheres, but very few useful particles are really homogeneous spheres. My concern here is towards the end of the paper where the discussion gets more speculative and wide ranging, that it goes too far at times.

R19: We acknowledge that the paper becomes more speculative towards its end, but we are proposing new research lines and experimental work that could further improve the way we interpret marine optical backscattering measurements and thus increase the potential of these observations for biogeochemical applications. We understand the Reviewer concerns and more experiments are first needed to test the hypothesis that from the coated-sphere model we can retrieve information on marine particle characteristics. For example, our preliminary results discussed in Lines 299-328 of the previous version of the manuscript made us suppose that particle characteristics can be deduced from a coated-sphere model. However, as we have already highlighted in the last paragraphs of the discussions, additional data and the use of complementary techniques are first needed to make progress towards this direction.

A19: Lines 322-333 (revised ms): we replaced “*could*” with “*might*”, and we added “*we hypothesise that*” to make explicit that we are speculating on the possibility to extract particle information from coated-sphere models.

Specific comments

Line 195: Not sure why the sentence starts with “in contrast to what Mie theory predicts” Mie theory predicts what it does for homogeneous spheres, coated spheres do something else....Sentence would make more sense without this phrase.

R20/A20: Lines 202-205 (revised ms): removed. We clarify that the term b_{bp} -to- c_p is equivalent to the backscattering ratio (i.e., b_{bp} -to- b_p where b_p is the total scattering) only when light absorption of particles is negligible: “*The b_{bp} -to- c_p ratio approximates, when the light absorption by particles is negligible (see methods), the ratio between b_{bp} and total scattering (i.e., backscattering ratio) which measures the efficiency with which light is backscattered by particles (Morel and Bricaud, 1986).*” We changed the caption of Figure 7 accordingly, and removed the reference Twardowski et al. (2001).

Line 205, figure 6a, I don’t understand this figure at all. It is the ratio of what?

R21: Figure 6a shows the median ratio between all particle size distributions collected after and before filtration through the 1- μ m filter. This ratio is the expression of the filtration efficiency as a function of particle size. The data presented in this figure allow us to quantify which submicron particles passed through the filter and thus show the reliability of the results plotted in Figure 6b.

A21: Legend Figure 6a now reads “*Ratio of particle size distributions measured after and before filtration of 5-m water samples trough a 1- μ m filter as a function of the particle diameter (D, units of μ m). Each point is the median of all ratios for samples collected during the AMT26 cruise (N=22; Supplementary Fig. 1) for a given particle diameter. Red dashed lines represent the first and third quartiles.*”; y-axis titles of Figure 6 have been updated.

Figure 6b, I think 40 % of the backscattering being less than 1 μ m is a fairly significant amount. Did you consider this when choosing parameters for the layered sphere (that you only had to come up with 60% of the backscattering?). Too bad the size fractionation experiment was not done more often, or that you had more data about the <1 μ m component.

R22: The results of the simulations in Figure 3 and the data shown in Figure 6b are entirely independent outcomes. More importantly, the data in Figure 6b come from water measurements before and after filtration to 1- μ m filter and no modelling has been applied to get these results (as specified in Line 185 of the previous version of the manuscript). Boxplots represent the ratios of optical measurements acquired by optical sensors before and after filtration. In addition, the main point emerging from Figure 6b is that the in-situ contribution of submicron particles to the backscattering signal, though fairly significant (40%), is only half the contribution expected by

considering a homogeneous sphere model (80%; Stramski and Kiefer, 1991). This is stated in Lines 227-247 of the previous version of the manuscript and Lines 238-241 of the revised manuscript.

Unfortunately, the size fractionation experiment could not be executed more than once per day during the AMT26 cruise. The main reason was that we needed also coincident particle size distribution (PSD) measurements to verify the efficiency of the filtration (i.e., Figure 6a), and the Coulter counter could be run only once a day for logistical reasons. Sample storage and utilization of preservatives was not recommendable to avoid any modification in particle characteristics and particle agglomeration from sampling to analysis. We also agree that more data below 1 μm would be desirable, but these data have been hard to measure throughout the history of ocean optics. We believe that the dataset we collected and presented in this manuscript is one of the few (if not the only one) in which coincident particle size distribution, c_p and b_{bp} data have been collected along a meridional transect in the Atlantic Ocean. Future work should of course expand upon the findings presented here.

A22: Line 189 (revised ms): we replaced “*small*” with “*smaller-than-expected*” contribution to optical backscattering. We adequately changed the brief title of caption of Figure 6.

Line 196 (revised ms): we now specify that is the “*measured*” b_{bp} signal.

Lines 258 to 271: this paragraph doesn’t really add anything to the paper or your case and is fairly obvious.

R23: We respectfully disagree with the Reviewer. We do consider it useful to clearly show what are the consequences for the interpretation of b_{bp} when modelling complex natural particles using the homogeneous-sphere model, especially because Nature Communications is a journal with a very diverse audience. We recognize the example in Figure 8 could be obvious for specialists in the field as the Reviewer is, but it can be instructive and helpful for scientists and students approaching marine optics and/or its biogeochemical applications.

A23: Lines 272-274 (revised ms): the sentence now reads “*To understand what might be the consequences of using the homogeneous-sphere model to simulate the scattering properties of complex natural particles (e.g., phytoplankton), it is instructive to attempt to model the scattering of a complex particle by only using homogeneous spheres.*”

Lines 285 -288. You first say that if particles are homogeneous spheres they will backscatter less...fine, that is the point of the paper. But then the next sentence says that this is because submicron particles backscatter more...these two sentences independently are fine (particularly if you take out “This is because”), but the reasoning doesn’t really flow at all..

R24/A24: Lines 299-300 (revised ms): the sentence now reads “*Some marine particles might, however, be represented by homogeneous models (Clavano et al., 2007; Xu et al., 2017) that could modify the contribution to the optical backscattering by the different size fractions.*”

Line 310-311. This statement seems to be very repetitive in the article. By now the reader should understand (and has read several times in the paper) that backscattering is sensitive to particle structural and compositional characteristics.

R25/A25: Line 321 (revised ms): the sentence has been removed.

Line 342-344. Are you trying to say that c_p and b_{bp} are “grossly” correlated with each other across large regions, but not in the fine scale (hence anti correlation in diel cycles)? The two statements, correlated and exhibiting different behaviors, seem conflicted.

R26: The interpretation is right and this is exactly what emerges from the current literature (cited in the ms). The possible reasons for this conflictual behaviour between c_p and b_{bp} have been discussed in previous papers, using exclusively the Mie theory (i.e., the homogeneous sphere). We thus think that the results of our manuscript may help opening the door to new hypotheses and experimental work as we have highlighted in Lines 344-347 of the previous version of the ms.

A26: Lines 354-357 (revised ms): The last two sentences of the paragraph have been modified and now read “*Understanding the contribution of phytoplankton to in-situ measurements of c_p and b_{bp} might thus increase our capability to predict oceanic carbon biogeochemical rates. We therefore suggest that a potential new avenue of research could be to identify what changes in cell ultrastructures and composition govern diel variations of optical scattering properties*”.

Line 369; coincidentally, a paper just came out in Applied Optics about coated spheres and why they are needed to explain polarization in the ocean.

R27/A27: Line 381-384 (revised ms): The paper by Tzabari et al. (Applied Optics, 2018) is now cited. The sentence now reads “*To characterise particle structural characteristics, integrated approaches based on microscopic and imaging analysis, coupled with optical measurements in situ and in the laboratory, such polarized light scattering (Quinby-Hunt et al., 1989; Tzabari et al., 2018), are needed.*”

Line 495: there is a step missing here. The way it sounds Scatnlay gives you VSF’s and then you have to integrate them to get Q_{bb}.

R28: Yes, the procedure has not been entirely described.

A28: Lines 501-572 (revised ms) we now report all the equations used to calculate Q_{bb} as well as the volume scattering functions and the elements of the Mueller matrix. The entire section now reads:

“Modelling particulate optical scattering properties. *The beam attenuation and optical backscattering coefficients of particles were modelled at the wavelength 532 nm as follows (Stramski and Kiefer, 1991):*

$$c_p(n) = \frac{\pi}{4} \int_{D_{min}}^{D_{max}} Q_c(D, n) D^2 F(D) dD \quad Eq. (1)$$

and

$$b_{bp}(n) = \frac{\pi}{4} \int_{D_{min}}^{D_{max}} Q_{bb}(D, n) D^2 F(D) dD \quad Eq. (2)$$

where D is the particle diameter (units of m) with minima and maxima as defined from PSD measurements (i.e., $(0.588-60) \cdot 10^{-6} m$ for AMT26 and $(1.4-42) \cdot 10^{-6} m$ for AMT22 samples). We did not extrapolate the PSDs for particles with diameters $< D_{min}$ that consequently were not modelled. For each individual particle, $Q_c(D, n)$ and $Q_{bb}(D, n)$ are the dimensionless efficiency factors for beam attenuation and optical backscattering respectively, at 532 nm, and the vector n represents the characteristics of the coated sphere with refractive indices (relative to that of seawater; Bricaud and Morel, 1986) of its core and coat, n_1 and n_2 , respectively and fractional thickness of the coat tk_2 ; the term $\pi D^2/4$ indicates the particle cross-sectional area (units of m^2); and $F(D)$ is the concentration of particles for a given diameter as obtained from the PSD measurements (units of particles m^{-4}). The values of n_2 were varied between 1.02 and 1.22 (Aas, 1996) while those of tk_2 varied between 2-30% of the radius of the particle. The volume-average refractive index (Bricaud et al., 1992) of the particle was forced to be equal to 1.06, which was the single value that best fitted the beam attenuation measurements through the homogeneous-sphere assumption (see results). As a consequence, the refractive index of the core n_1 varied as a function of n_2 and tk_2 . A total of 609 combinations of tk_2 and n_2 were tested with the coated-sphere model. Homogeneous spheres were represented as a coated sphere with $tk_2=0$ and n_1 values between 1.02 and 1.22 (Aas, 1996). Particles counted within each size bin of the measured PSD were considered as monodispersed and non-absorbing populations at the selected wavelength (Morel and Bricaud, 1986). We expect the results of this modelling analysis to be more sensitive to the structural complexity of the spheres than to the impact of light absorbing material on b_{bp} (Bricaud et al., 1992; Zaneveld and Kitchen, 1995), and thus to be representative also of other portions of the visible spectrum.

Look-Up-Tables of $Q_c(D, n)$ for homogeneous and coated spheres were generated by using the “Scatnlay” software (Peña and Pal, 2009), for angles between 0 and 180° (0.1° increments) and logarithmically-spaced equivalent spherical diameters between 0.4 and 70 μm . The $Q_c(D, n)$ values both for homogeneous and coated spheres were corrected for the acceptance angle effect to be consistent with WETLabs ac-9 and ac-s in-situ measurements (Boss et al., 2009).

The values of $Q_{bb}(D, n)$ were computed as:

$$Q_{bb}(D, n) = Q_b(D, n) \frac{\int_{\pi/2}^{\pi} S_{11}(\theta, D, n) \sin \theta d\theta}{\int_0^{\pi} S_{11}(\theta, D, n) \sin \theta d\theta} \quad \text{Eq. (3)}$$

where Q_b is the dimensionless efficiency factor for the total scattering (output of the freely available “Scatlay” software; Peña and Pal, 2009) and $S_{11}(\theta, D, n)$ is the top left component (i.e., 11) of the scattering matrix computed as described below. To compute the volume scattering function and the other non-normalized components of the Mueller matrix we proceeded as follows (Peña and Pal, 2009). First, for each set of particle parameters (i.e., D and n for the wavelength 532 nm) we computed the following elements of the scattering matrix:

$$S_{11}(\theta, D, n) = 0.5[|S_2(\theta, D, n)|^2 + |S_1(\theta, D, n)|^2] \quad \text{Eq. (4)}$$

$$S_{12}(\theta, D, n) = 0.5[|S_2(\theta, D, n)|^2 - |S_1(\theta, D, n)|^2] \quad \text{Eq. (5)}$$

$$S_{33}(\theta, D, n) = 0.5[S_1(\theta, D, n)S_2^*(\theta, D, n) + S_1^*(\theta, D, n)S_2(\theta, D, n)] \quad \text{Eq. (6)}$$

$$S_{34}(\theta, D, n) = 0.5i[S_1(\theta, D, n)S_2^*(\theta, D, n) - S_1^*(\theta, D, n)S_2(\theta, D, n)] \quad \text{Eq. (7)}$$

where $S_1(\theta, D, n)$ and $S_2(\theta, D, n)$ are the complex scattering amplitude outputs of the “Scatlay” software (Peña and Pal, 2009) and the starred quantities indicate complex conjugates. Each $S_{ij}(\theta, D, n)$ component was then normalized by the integral of the $S_{11}(\theta, D, n)$ component to compute normalized elements of the scattering matrix:

$$\overline{S_{ij}}(\theta, D, n) = \frac{S_{ij}(\theta, D, n)}{2\pi \int_0^{\pi} S_{11}(\theta, D, n) \sin \theta d\theta} \quad \text{Eq. (8)}$$

We then computed the Mueller matrix components of particle populations composed of modelled coated spheres and the measured particle size distributions, $F(D)$, as:

$$M_{ij}(\theta, n) = \frac{\pi}{4} \int_{D_{\min}}^{D_{\max}} \overline{S_{ij}}(\theta, D, n) Q_b(D, n) D^2 F(D) dD \quad \text{Eq. (9)}$$

with $M_{11}(\theta, n)$ resulting in the modelled volume scattering functions.

To compare the shape of the volume scattering functions with existing measurements, we normalized $M_{11}(\theta, n)$ by its value integrated between 0 and π (i.e. total scattering) to obtain the corresponding phase functions as:

$$\text{Normalized VSF}(\theta, n) = \frac{M_{11}(\theta, n)}{2\pi \int_0^{\pi} M_{11}(\theta, n) \sin \theta d\theta} \quad \text{Eq. (10)}$$

Finally, to compare the polarized components to existing datasets, we normalized each component of the scattering matrix by $M_{11}(\theta, n)$ as follows:

$$P_{ij}(\theta, n) = \frac{M_{ij}(\theta, n)}{M_{11}(\theta, n)} \quad \text{Eq. (11)}$$

where now i and j are not simultaneously equal to 1.”
Equations 13 and 14 have been modified accordingly.

Additional minor changes to the manuscript:

Line 32 (revised ms): changed “atmospheric organic carbon” to “atmospheric CO₂”.

Line 36 (revised ms): the sentence now reads “*To achieve this understanding, measurements of the light scattered by marine particles are crucial.*”

Lines 50-52 (revised ms): the reference Kostadinov et al. (2009) has been removed because it was redundant with respect to Kostadinov et al. (2016; Reference#21 in the revised ms). The sentence now reads: “*The b_{bp} spectrum provides a proxy of the particle size (Slade and Boss, 2015) and can be used to analyse the phytoplankton community structure and derive group-specific carbon biomass (Kostadinov et al. 2016).*”

Lines 373-374 (revised ms): the following sentence has been added “*Finally, how the modelled coated spheres relate to oceanic detrital particles remains to be understood.*”

Lines 396 (revised ms): the reference Stramski & Wozniak (2005) has been removed because it was redundant with respect to Stramski et al. (2004; Reference#32 in the revised ms).

Line 483 (revised ms): the reference Slade et al. (2010) has been replaced by Dall’Olmo et al. (2012) to meet the total number of references required by the journal. The sentence has been modified accordingly.

Lines 576-577 (revised ms): the sentence now reads “*The model performance was evaluated by calculating the systematic error, in logarithmic space, as follows*”. The reference Campbell (1995) has been removed.

A few small adjustments on the text were made to further improve the manuscript. All other changes (e.g., section and subheading titles, abstract length) have been made to agree with Nature Communications policies and the checklist provided by the Editorial team.

REVIEWERS' COMMENTS:

Reviewer #1 (Remarks to the Author):

I appreciate the detailed responses to my comments and those of the other reviewers provided by the authors and the associated revisions in the current version of the manuscript. I feel the authors have adequately addressed my concerns and now recommend publication.

Reviewer #2 (Remarks to the Author):

The authors have adequately addressed all my comments and concerns both within their response and within the revised manuscript and I am thus supportive of publication of the new version.

Reviewer #3 (Remarks to the Author):

They have addressed my concerns sufficiently.

Revision of the manuscript

The open-ocean missing backscattering is in the structural complexity of particles
by Organelli et al.

We sincerely thank the three Reviewers for the time they invested in reviewing our manuscript and for recommending publication of our revised version as it is. Hereafter, the reviewer's comments appear in black.

Reviewer's comments:

Reviewer #1 (Remarks to the Author):

I appreciate the detailed responses to my comments and those of the other reviewers provided by the authors and the associated revisions in the current version of the manuscript. I feel the authors have adequately addressed my concerns and now recommend publication.

Reviewer #2 (Remarks to the Author):

The authors have adequately addressed all my comments and concerns both within their response and within the revised manuscript and I am thus supportive of publication of the new version.

Reviewer #3 (Remarks to the Author):

They have addressed my concerns sufficiently.